# Circulating HPV DNA as a Biomarker for Pre-Invasive and Early Invasive Cervical Cancer: A Feasibility Study

**DOI:** 10.3390/cancers15092590

**Published:** 2023-05-02

**Authors:** Stacey J. Bryan, Jen Lee, Richard Gunu, Allison Jones, Adeola Olaitan, Adam N. Rosenthal, Ros J. Cutts, Isaac Garcia-Murillas, Nick Turner, Susan Lalondrelle, Shreerang A. Bhide

**Affiliations:** 1UCL Elizabeth Garrett Anderson Institute for Women’s Health, Faculty of Population Health Sciences, University College London, Medical School Building, 74 Huntley Street, London WC1E 6AU, UK; 2The Institute of Cancer Research, Fulham Road, London SW3 6JB, UKros.cutts@icr.ac.uk (R.J.C.); isaac.garcia-murillas@icr.ac.uk (I.G.-M.); nick.turner@icr.ac.uk (N.T.);; 3The Royal Marsden Hospital, Fulham Road, London SW3 6JJ, UK; 4Translational Research Lab, Department of Women’s Cancer, IfWH, Ground Floor POGB, 72 Huntley Street, London WC1E 6DD, UK

**Keywords:** human papillomavirus, cervical cancer, next generation sequencing, liquid biopsy, plasma, circulating DNA

## Abstract

**Simple Summary:**

Over 95% of cervical cancers are caused by the human papilloma virus (HPV). Locally advanced HPV-related cancers release tiny bits of tumour containing HPV-DNA into the blood (cHPV-DNA). The presence of cHPV-DNA in the blood can serve as a potential marker of cancer and should be absent in pre-cancerous conditions (cervical intraepithelial neoplasia (CIN). We have developed an ultra-sensitive and specific test to measure blood cHPV-DNA levels. We performed a feasibility study to confirm our expectation that cHPV-DNA is not found in the blood of patients with CIN but is found in the blood of patients with very early cancers. In this study, we have confirmed that cHPV-DNA is absent in the plasma of women with CIN. In early cervical tumours, there was a low detection rate of cHPV-DNA. Therefore, more sensitive tests are required before cHPV-DNA can be used for the detection of very early cervical cancers.

**Abstract:**

Background: High-risk HPV infection is responsible for >99% of cervix cancers (CC). In persistent infections that lead to cancer, the tumour breaches the basement membrane, releasing HPV-DNA into the bloodstream (cHPV-DNA). A next-generation sequencing assay (NGS) for detection of plasma HPV circulating DNA (cHPV-DNA) has demonstrated high sensitivity and specificity in patients with locally advanced cervix cancers. We hypothesised that cHPV-DNA is detectable in early invasive cervical cancers but not in pre-invasive lesions (CIN). Methods: Blood samples were collected from patients with CIN (*n* = 52) and FIGO stage 1A-1B CC (*n* = 12) prior to treatment and at follow-up. DNA extraction from plasma, followed by NGS, was used for the detection of cHPV-DNA. Results: None of the patients with pre-invasive lesions were positive for CHPV-DNA. In invasive tumours, plasma from one patient (10%) reached the threshold of positivity for cHPV-DNA in plasma. Conclusion: Low detection of cHPV-DNA in early CC may be explained by small tumour size, poorer access to lymphatics and circulation, and therefore little shedding of cHPV-DNA in plasma at detectable levels. The detection rate of cHPV-DNA in patients with early invasive cervix cancer using even the most sensitive of currently available technologies lacks adequate sensitivity for clinical utility.

## 1. Background

In the UK, cervical cancer is the 14th most common cancer in females [1]. Overall five-year survival is approximately 61% but is dependent on the stage at diagnosis, with 96% of women surviving the disease for a year or more if diagnosed early, versus 50% who are diagnosed at a later stage [1]. Equally, age at diagnosis plays a role, with around 90% of women diagnosed between ages 15 and 39 surviving for 5 years or more, compared to 25% of those aged 80 or over.

Risk factors for cervical cancer are multiple [2], including smoking [3], sexual practices [4], and exposure to and persistence of high-risk human papillomavirus (HRHPV)—which is detected in 99.7% of cervical cancers and has consequently been listed as a carcinogen by the International Agency for Research on Cancer (IARC) [1,5].

Many cancers are known to shed fragments of DNA (circulating tumour DNA, ctDNA) into the bloodstream. Most cervical cancers are causally related to the human papilloma virus (HPV) [6]. In the tumour, HPV DNA is integrated into the host genome or present in episomal form [7] and can be released into the bloodstream. Thus, circulating HPV-DNA (cHPV-DNA) can potentially be used as a detection marker for HPV-related cervical cancer.

Various studies investigating cHPV-DNA as a tumour marker in cervical cancer patients have demonstrated between 11% and 45% positivity rates of cHPV-DNA in patients with invasive cervical cancer at diagnosis, suggesting a low sensitivity of the PCR assays [8,9,10].

cHPV-DNA has also been detected in HPV-positive head and neck cancers, with many of the studies employing PCR methods, which have a sensitivity of 19–79% in locally advanced diseases. The use of digital droplet PCR (ddPCR) using probes for HPV 16 and 33 has been shown to increase the detection of HPV ctDNA in oropharyngeal squamous cell carcinomas (OPSCC) to almost 96% [11]. Similarly, improved detection using ddPCR has been described in cervical cancer in the metastatic setting [12]. There has been limited success in detecting cHPV-DNA for cervical cancer diagnosis due to the lack of sensitivity of conventional PCR assays [13].

Traditionally, HPV genotyping involved the amplification of a region of the L1 gene with next-generation sequencing (NGS) using specific primers. Lee et al. developed an amplicon-based next-generation sequencing assay to detect circulating plasma viral DNA (‘*HPV16-detect*’) to assess the response to chemoradiotherapy in locally advanced head and neck cancer [14]. This novel NGS assay comprises a 39-amplicon single pool panel that covers 34 distinct regions of the HPV 16 genome, as well as five amplicons of human reference genes. In their cohort, the assay demonstrated 100% sensitivity and 93% specificity in the detection of HPV 16-positive head and neck cancers at diagnosis. In addition, by tracking HPV through and post-chemoradiation, they were able to predict response and residual disease, demonstrating that cHPV-DNA can be used as a marker of response to treatment.

In their other study, which included a cohort of HPV-related anal squamous cell carcinoma patients post-chemoradiotherapy, Lee et al. expanded the novel NGS assay to include HPV subtypes 16, 18, 31, 33, 35, 45, 52, and 58 (*panHPV-detect*) [15]. This included two primer pools covering distinct regions (single nucleotide polymorphisms) of the HRHPV genome to detect cHPV-DNA. In this study, they demonstrated 100% sensitivity and specificity of *panHPV-detect* in diagnosing the high-risk HPV subtypes prior to treatment of anal squamous cell carcinoma. Furthermore, *panHPV-detect* was able to identify those patients who either had residual disease post-treatment or relapsed a few months later [15].

There has been some work surrounding early detection and follow-up of treated cervical disease, with circulating DNA playing a promising role. However, these studies tend to include advanced cancers rather than early invasive cancers. In this prospective cohort study, we aimed to investigate the use of this highly sensitive assay—*panHPV-detect*—for the detection of cHPV-DNA in both pre-invasive cervical lesions and early invasive cervical cancer. As HPV does not enter the bloodstream in its infective phase, we hypothesized that cHPV-DNA would not be detected in pre-invasive cervical disease but would be detectable in early-stage cancers.

## 2. Methods

This study was approved by the institutional board (Ref. no. CCR 4157) and ethics committee (Ref. no. 14/NE/1055). Patients who were referred to the colposcopy clinic at UCLH with cervical smears showing high-grade dyskaryosis or persistent mild dyskaryosis and HPV-positives, and who were attending for cervical treatment in the form of Large Loop Excision of the Transformation Zone (LLETZ) or knife cone biopsy, were recruited. Inclusion criteria were those over the age of 18, referred with HSIL and HPV^+^ve, and able to give informed consent. The exclusion criteria were those with a previous diagnosis of cancer, previous CIN treatment, and other HPV-associated lesions (VIN/VaIN/AIN). Following written, informed consent, 20 mL of venous blood was collected in EDTA tubes prior to treatment and then at the next appointment following treatment to assess for cHPV-DNA. The histological findings were correlated with the *panHPV-detect* result. Women who could not tolerate treatment in the outpatient setting or were undergoing general anesthesia for treatment of a larger lesion were seen in the clinic at the time of consent for their surgery and recruited to this study. A venous blood sample was then taken on the morning of surgery.

A second cohort of patients with biopsy-proven early-stage (FIGO 2018 1A/1B) invasive disease was recruited from the UCLH Gynaecological Oncology clinic., in London. These patients were recruited to this study in the outpatient clinic based on radiological staging and prior to definitive treatment, which was typically a simple or radical hysterectomy.

### 2.1. Sample Size Calculation

In the head and neck cancer pilot study, *HPV16-detect* demonstrated 100% sensitivity and 93% specificity in detecting HPV16-positive cancers at diagnosis. To have 80% power to detect a true sensitivity of 99%, we calculated that at least 19 HPV-positive, invasive cancer patients needed to be recruited. As approximately 95% of cervical cancer patients are HPV-positive, we aimed to recruit 22 patients with early invasive carcinoma. To confirm 93% specificity for pre-invasive disease, we required 37 patients with high-grade squamous intraepithelial lesions (HSIL) to be recruited.

### 2.2. Sample Collection and Processing

Whole blood sampling and plasma extraction were conducted at the Translational Research Laboratory at UCL, London, UK. The frozen plasma samples were then transported to the Chester Beatty Lab at the Institute of Cancer Research (ICR), London, UK for further analysis.

Additionally, 20 mL of blood was centrifuged at 1600× *g* for 10 min within 48 h of collection. The resulting plasma was then centrifuged again at 1600× *g* for 10 min to remove any cell sediment, aliquoted, and frozen at −80 °C. DNA was extracted from 5 mL of plasma using the QIAamp Circulating Nucleic Acid Kit (Qiagen) according to the manufacturer’s instructions. DNA was eluted in 50 μL of AVE buffer and stored at −20 °C. Plasma DNA was quantified using a Bio-Rad QX200 ddPCR system (Bio-Rad, Hertfordshire, UK), using ribonuclease P (RNase P) as a reference gene, as previously described [16].

We performed an initial validation of panHPV-detect by testing its ability to correctly identify the HPV subtype in cervical cancer tissue samples. We obtained tumour DNA extracted from cervical cancer tissue previously typed for the HPV subtype using polymerase chain reaction (RT-PCR) for E6 and E7 mRNA using validated assays. Five samples for each of the eight high-risk HPV subtypes were obtained from the Scottish HPV archive. Samples from five HPV-negative patients with head and neck cancer were recruited in the head and neck cancer cohort of this study, and 19 patients with breast cancer enrolled in the prospective sample collection study (PlasmaDNA, CCR3297, REC Ref No: 10/H0805/50) were used as negative controls. HPV status was confirmed as negative with E7 RT-PCR performed on RNA extracted from the samples. None of the negative controls were known to have pre-cancerous lesions. Written informed consent was obtained from all participants. Following the initial validation, we investigated whether panHPV-detect was able to detect cHPV-DNA in the pre-treatment plasma of patients recruited in this study.

To classify HPV^+^ and HPV^−^ samples using panHPV-detect in tissue, we set a threshold whereby a sample was classified positive if there were 10 reads present from more than 30% of different HPV amplicons for each subtype. To assess the threshold for the number of amplicons needed for a positive panHPV-detect readout in plasma, a ROC analysis was used. In the first step, HPV status and subtype were assigned in tissue using E7 mRNA to separate the two groups. The number of amplicons with greater than 10 reads at baseline was inputted for each patient to find a suitable threshold for this parameter. Sorting the values in both HPV^+^ and negative groups and averaging adjacent values in the sorted list generated a list of thresholds. Based on the ROC analysis, a threshold that gave the greatest sensitivity and specificity for each HPV subtype was selected as the threshold for the classification of plasma as HPV DNA-positive. Based on the ROC analysis, a threshold of 7 amplicons with more than 10 reads were the thresholds that gave the greatest sensitivity and specificity and were selected as the thresholds for the classification of plasma as HPV DNA-positive.

DNA library preparation was performed, and then Ion Torrent qPCR Quantitation with final sequencing was performed at the Tumour Profiling Unit of the ICR. Ion Torrent Libraries were prepared using Ion Ampliseq Library Preparation kit 2.0 (Thermo Fisher Scientific, Waltham, MA, USA) according to the manufacturer’s instructions using 5 ng of tissue DNA or 3 ng of plasma DNA per primer pool, except the volumes of the Ion Ampliseq Library Preparation kit and the Custom Primer Pools have been reduced by half. Reads were aligned to an amalgamated reference containing scaffolds for each of the HPV genotypes as well as the reference human targeting genes using default settings in module map4 of the TMAP on the Ion Torrent machine. Bedtools v2.23.0 (Salt Lake City, UT, USA) [17] was used to extract on-target reads from the aligned files with a minimum overlap of 90% with amplicons in the panel. Additionally, reads with a mapping quality of <30 were removed using Samtools v1.2 (Hinxton, UK) [18]. Reads were split into those covering human and HPV amplicons, and coverage of each portion and each genotype was calculated individually.

In order to classify HPV^+^ and HPV^−^ samples using *panHPV-detect* in tissue, we set a threshold whereby a sample was classified positive if there were 10 reads present from more than 7 different HPV amplicons for each subtype.

### 2.3. Cost of Sample Analysis

The cost of sample analysis was £132 per sample (Appendix A). The bulk cost of the sample analysis is apportioned to the sequencing costs and the library preparation kit costs. We have optimized the library preparation protocol to use half the recommended volume of reagents for library preparation.

## 3. Results

We recruited 52 patients between November 2018 and January 2020. The COVID-19 pandemic necessitated the suspension of study recruitment and sample collection, including follow-up of already recruited patients, in January 2020. All non-urgent clinical studies were placed on hold as laboratory services and resources were diverted to the pandemic. In addition, there was a cessation of the cervical screening programme for 2 months, a restriction on colposcopy appointments, and the introduction of social distancing rules, which meant that further recruitment from and follow-up at clinics were halted.

None of the recruited patients had a history of prior HPV vaccination. Patient demographics can be found in Table 1.

### 3.1. Detection of CHPV-DNA in Baseline Samples

Of the 52 patients recruited, 12 had early invasive cervical cancer and 40 had pre-invasive disease (Table 2). The mean ages of participants with cancer and pre-invasive disease were 39 and 35 years, respectively. At the time of baseline blood collection, the final histology was not known; therefore, all samples were included for analysis. Analysis was performed blind to the referral HPV status. An adequate quantity of DNA could be extracted from 50 of the 52 patients, with two samples (9 and 11) failing the library preparation stage due to insufficient volume of DNA despite PCR amplification.

Using the pre-set threshold of 10 reads present from >7 HPV amplicons, only 2 of the 50 baseline samples were HPV positive. HPV reads were detected in 15 other samples but in no more than 6 amplicons per HPV subtype and therefore did not reach the threshold for positivity (Table 3).

### 3.2. Detection of cHPV-DNA in Early Invasive Cervical Cancer

Final histology was available after all plasma samples were analysed and DNA sequencing performed (including follow-up samples). Out of the 12 cancer samples, 2 were found to be at a more advanced stage (FIGO 2B) than the inclusion criteria and therefore had to be excluded from this study (1 of these reached the threshold for positivity for HPV). Of the remaining 10 samples, only one was positive for HPV as per threshold criteria. This sample came from a patient who had stage 1A1 cervical cancer and was found to be HPV 18-positive, which correlated with the HPV type on referral cytology. This patient was an ex-smoker, under 30 years old, and had a small tumour. The other five patients whose histology indicated FIGO stage 1A1 tumours, did not reach the threshold for HPV positivity in plasma. None of the four 1B1 tumours had HPV in the plasma at levels that reached the threshold for positivity.

### 3.3. Detection of cHPV-DNA in Pre-Invasive Cervical Disease

None of the plasma samples from patients with final histology showing CIN 1–3 or cervical glandular intraepithelial neoplasia (CGIN) reached the threshold for positivity for HPV (Table 3). In 10 of these samples, HPV amplicons were detected, but at numbers far less than the threshold. Combining the results from the detection of CHPV-DNA in pre-invasive and invasive disease gives a sensitivity of 10% (95% CI, 0.24–44.5%) and a specificity of 100% (95% CI, 91.2–100%). There was no correlation between the number of amplicons detected and the grade of CIN or stage of cancer.

### 3.4. Detection of cHPV-DNA in Follow-Up Samples

Only eighteen patients out of the cohort of 50 were followed up prior to the COVID-19 pandemic. Two of the patients (9 and 11) had to have follow-up samples analysed despite baseline samples not being available (all samples were blinded and run on the same chip, regardless of follow-up or baseline). It was not known at the time of the blood draw that the baseline plasma had failed the PCR and library preparation stages. Of these 18 follow-up samples, 10 had HPV reads detected but none at >6 amplicons, and therefore none of the follow-up samples reached thresholds for positivity (Table 4). Sample number 25, which was the only positive baseline sample—detecting HPV 18 prior to treatment—was negative at follow-up.

Of the 10 samples that had HPV reads at follow-up, only 2 correlated with the baseline HPV read—in one case, there was no baseline sample to compare. Seven samples detected HPV types that were either new or did not correlate with the baseline HPV types (Table 4).

## 4. Discussion

The aim of this project was to determine whether *panHPV-detect* could detect cHPV-DNA in the plasma of patients with early-stage cervical cancers, as it has previously been shown to have high sensitivity and specificity for locally advanced HPV-associated cervical, head and neck, and anal cancers [14,15]. Using the thresholds of positivity for head and neck and anal cancers, this study has demonstrated that none of the patients with pre-invasive disease had cHPV-DNA in their plasma (specificity 100%), which is compatible with the first of our study hypotheses: that pre-invasive disease does not shed detectable levels of cHPV-DNA into the circulation. However, we demonstrated an assay sensitivity of only 10% for early-stage invasive disease.

The low sensitivity of this study correlates with other similar studies. Pornthanakasem et al. investigated the diagnostic and prognostic potential of circulating cHPV-DNA in patients with cervical cancer [9]. They collected tissue from 63 patients with cervical cancer and obtained blood samples from the same cohort and from 20 healthy blood donors. Only 6/50 patients with HPV-associated cervical cancer had circulating DNA, indicating low sensitivity. However, circulating DNA was not found to be present in normal controls or in HPV-negative cervical cancer, indicating a high specificity. Furthermore, the HPV genomes from both the tumour and the plasma matched, indicating that the circulating viral DNA was derived from the tumour cells. This study also reported that HPV in the plasma was associated with distant metastases, recurrence within 1 year, and therefore a poor prognosis [9].

Sathish et al. also looked at cHPV-DNA in cervical cancer patients. They collected cervical biopsies and plasma samples from 58 women with invasive cervical cancer. In addition, they also collected samples from 10 women with the pre-invasive disease and 30 control women matched for age. All paired samples for the control group were negative for HPV DNA; however, only eight (11.8%) were positive for cHPV-DNA in the cervical cancer group. In the CIN group, all 10 patients were positive for HPV in cervical biopsies, but none were positive in plasma samples. Of the eight patients who were plasma DNA positive, seven were HPV 16-positive, and six were at stage 3–4 cancer. Cervical tissue and plasma pairs were correlated on sequencing [10].

Both the aforementioned studies used more conventional PCR techniques rather than the novel NGS assay used in this study, which we had expected to achieve greater sensitivity for the detection of disease.

A systematic review by Trigg et al. investigated the possible factors that influence the quality and yield of circulating free DNA [19]. The factors evaluated included the specimen type, collection tube type, time to processing, centrifugation protocols, and methods of cfDNA isolation and quantification. Based on the results of this systematic review, our protocols followed the recommendations suggested to maximise the yield of DNA from plasma. Of the two samples with insufficient DNA, the extractions were repeated but gave the same results, suggesting a true deficit in cfDNA in those samples rather than an error in technique.

We have carried out a similar study [20], which evaluated *panHPV-detect* in locally advanced cervical cancers in patients undergoing chemoradiotherapy. This study has demonstrated high sensitivity (88%) and specificity (100%) for our NGS assay in detecting cHPV-DNA at baseline and following treatment completion. This suggests that the stage or volume of tumour plays a role in the amount of circulating tumour DNA that may be released into the plasma.

Gu et al. published a recent meta-analysis of cHPV-DNA as a biomarker for cervical cancer [21]. They found 10 studies published between 2001 and 2018 that met their eligibility criteria and covered 684 patients with cervical cancer. Patients with primary or metastatic cervical cancers at stages 1–4 who were receiving all modalities of treatment were included. Tumour types were both squamous and adenocarcinomas, with a greater proportion being squamous cell tumours. None of the control or pre-invasive disease groups had positive cHPV-DNA in the serum. The overall pooled sensitivity and specificity were 0.27 (95% CI 0.24–0.30) and 0.94 (95% CI 0.92–0.96), respectively [22].

On further analysis of each study, the detection of cHPV-DNA in plasma in early-stage cervical cancer showed mixed results, with two studies showing no detection at stage 1B [9,23], three studies with <10% detection at stage 1B [10,13,24], and another three studies showing detection between 24 and 100% [12,22,25]. The studies that described high levels of cHPV-DNA in stage 1 tumours also noted that the viral DNA load was directly proportional to the clinical stage; the patients also tended to be high-risk, i.e., lymphovascular space invasion (LVSI), deep stromal invasion, and tumour size > 20 mm. This may go some way to explaining why many of the early-stage cancers in our study were negative for cHPV-DNA.

Fiala et al. described the challenges of using ctDNA for early-stage cancer diagnosis [26]. There is usually not enough ctDNA present in such cancers to allow diagnosis, which makes it difficult to design highly sensitive assays. They estimated that a tumour of 1 cm^3^, or 12.5 mm in diameter, will release enough ctDNA to represent 0.01% of all circulating DNA [27,28]. They concluded, therefore, that ctDNA could be used to detect tumours of approximately 10 mm but not smaller.

The strength of this study is that it focused on early-stage cervical cancers only. The low sensitivity for the detection of cHPV-DNA in early-stage cervical cancer may be explained by not only the small size of the tumour but also the inherent difficulties in the design of a sufficiently accurate HPV assay. The high specificity, however, does give some assurance that a positive test is more likely to be a true positive.

The limitation of this study is that we were unable to recruit the desired number of patients, particularly in the early cervical cancer cohort, due to the coronavirus pandemic, which also made follow-up difficult. The number did not meet the requirements of the power calculation. However, as only one cervical cancer sample was positive for cHPV-DNA, this likely would not have made a statistical difference. Another inherent limitation of this prospective cohort study was the difficulty of follow-up post-treatment. Again, as all but one of the baseline samples were negative, follow-up of these patients is unlikely to have added a significant finding to this study. Of the patients that were followed up, there were no discrepancies between baseline and follow-up samples, i.e., none became positive after treatment having had a negative baseline result. Most importantly, the sole positive patient was followed up and found to be negative after treatment.

## 5. Conclusions

Based on this study, we cannot recommend the use of *HPVdetect* as a biomarker for the diagnosis of early-stage cervical cancers, given the low sensitivity demonstrated. However, we have shown that circulating HPV is not detectable in pre-invasive disease, supporting the hypothesis that such lesions do not shed DNA into the circulation. Patients with pre-invasive disease have an effective screening tool based on high-risk HPV detection and cytology in cervical samples. Whilst work is ongoing on the use of *HPVdetect* as a potential biomarker for monitoring treatment effectiveness and predicting recurrence in locally advanced cervical cancers, further research is needed to establish whether any cDNA assay has sufficient sensitivity for reliable detection of early-stage cervical cancer.

## Figures and Tables

**Table 1 cancers-15-02590-t001:** Patient demographics.

Demographics	
**Age**	N (%)
25–35	34 (65)
36–45	13 (25)
46–55	3 (6)
>55	2 (4)
**Ethnicity**	
White British	32 (62)
European	9 (17)
Asian	5 (10)
Black	2 (4)
Other	4 (7)
**Smoker**	
Previous	11 (21)
Current	16 (31)
No	25 (48)
**HPV Subtype**	
16	8 (40)
18	3 (15)
Non 16/18	9 (45)

**Table 2 cancers-15-02590-t002:** Stage of cancer and grade of pre-invasive lesion.

Results	Stage	Number of Patients	Total
**Cancers**	1A1	6	12
	1B1	4	
	2B	2	
**Pre-invasive**	CIN 1	9	40
	CIN 2	17	
	CIN 3	11	
	CGIN	3	

**Table 3 cancers-15-02590-t003:** Stage of cancer and grade of pre-invasive lesion, with HPV subtype (where known) and number of amplicons read. * Excluded from study.

Sample No.	Diagnosis	HPV Type Cytology	HPV Type Plasma	Number of Amplicons > 6	Pos/Neg
42	1A1	-	16	1	NEG
44	1A1	-	52	1	NEG
52	1A1	-	-	0	NEG
24	1A1	16	-	0	NEG
25	1A1	18	18	11	POS
35	1A1	16	16	1	NEG
41	1B1	-	-	0	NEG
10	1B1	-	16	1	NEG
23	1B1	-	31	2	NEG
2	2B	18	18	79	POS *
4	2B	NON 16/18	-	0	NEG *
5	CGIN	-	16	1	NEG
31	CGIN	-	-	0	NEG
40	CGIN	-	-	0	NEG
49	CIN 1	16	-	0	NEG
1	CIN 1	-	-	0	NEG
8	CIN 1	NON 16/18	16	3	NEG
17	CIN 1	18	33	1	NEG
21	CIN 1	-	-	0	NEG
22	CIN 1	-	-	0	NEG
28	CIN 1	16	-	0	NEG
37	CIN 1	-	-	0	NEG
38	CIN 1	-	-	0	NEG
43	CIN 2	-	-	0	NEG
45	CIN 2	-	-	0	NEG
46	CIN 2	-	16	1	NEG
47	CIN 2	-	-	0	NEG
51	CIN 2	-	-	0	NEG
3	CIN 2	NON 16/18	-	0	NEG
6	CIN 2	NON 16/18	-	0	NEG
7	CIN 2	16	-	0	NEG
12	CIN 2	NON 16/18	-	0	NEG
18	CIN 2	-	-	0	NEG
19	CIN 2	-	16	2	NEG
20	CIN 2	NON 16/18	-	0	NEG
29	CIN 2	NON 16/18	-	0	NEG
34	CIN 2	-	16	1	NEG
36	CIN 2	-	33	0	NEG
39	CIN 2	-	-	0	NEG
48	CIN 3	-	-	0	NEG
50	CIN 3	-	-	0	NEG
13	CIN 3	16	16	1	NEG
14	CIN 3	16	-	0	NEG
15	CIN 3	16	-	0	NEG
16	CIN 3	-	-	0	NEG
26	CIN 3	-	-	0	NEG
27	CIN 3	-	16	0	NEG
30	CIN 3	-	-	0	NEG
32	CIN 3	-	-	0	NEG
33	CIN 3	-	16	0	NEG

**Table 4 cancers-15-02590-t004:** Follow-up sample HPV data.

Sample No.	Initial Diagnosis	HPV Type Plasma	HPV Type f/u	Number of Amplicons > 6	Pos/Neg
44	1A1	52	-	0	NEG
3	CIN 2	Non 16/18	16	2	NEG
5	CGIN	16	-	0	NEG
6	CIN 2	-	16	1	NEG
7	CIN 2	-	-	0	NEG
8	CIN 1	16	16/18	3	NEG
9	1B1	No sample	18	3	NEG
10	1B1	16	16	1	NEG
11	CIN 2	No sample	-	0	NEG
12	CIN 2	-	-	0	NEG
13	CIN 3	16	-	0	NEG
16	CIN 3	-	33	1	NEG
18	CIN 2	-	33	1	NEG
21	CIN 1	-	-	0	NEG
23	1B1	31	18	1	NEG
24	1A1	-	18	1	NEG
25	1A1	18	16	1	NEG
31	CGIN	-	-	0	NEG

## Data Availability

Data are contained within the article.

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
