# Peer review of "Circulating HPV DNA as a Biomarker for Pre-Invasive and Early Invasive Cervical Cancer: A Feasibility Study"

_cancers, 2023, doi:10.3390/cancers15092590_

Round 1

Reviewer 1 Report

Circulating DNA of human papilloma virus (cHPV-DNA) has potential for diagnosis of cervical cancers, most of which are caused by variants of HPV infection. However, tests using qPCR to detect cHPV-DNA in plasma have shown relatively poor sensitivities, likely due to poor sensitivity of conventional qRT-PCR assays. Use of NGS based methods of detection for cHPV (panHPV-detect) in plasma have produced results with better sensitivity for diagnostic as well as therapeutic biomarker applications. In this study, authors have used ion torrent DNA sequencing following the panHPV-detect-specific library preparation to test the hypothesis that pan-HPV-detect can classify early-stage cervical cancers but not pre-invasive cervical disease.

This manuscript is well drafted, easy to follow and presents some interesting results. However, I have several questions that I would like the authors to address –

Major Comments

 1. As the authors have mentioned in the discussion, this study suffers from lack of a balanced design. I believe this is a major drawback and it is difficult to truly judge the merit of this diagnostic method based on only 10-12 early-stage cervical cancer patients and nearly four times as many early invasive disease patients. I highly recommend recruiting more patients to your study for a more balanced design

2. Why did the authors use 3-5 ng of DNA when the kit recommends 10 ng of DNA.

Would there be a difference if larger amount of starting material was used for library preparation?

3. Did authors compare results with different alignment software? Authors should compare alignment with a few aligners, as sequence identification can be different between bwa vs star alone vs star+bowtie.

4. What is the justification behind ‘>10 reads of atleast 7 or more amplicons’ as the positivity classifier?

5. In Methods, author mention ‘using 5ng of tissue DNA or 3ng of  plasma DNA per primer pool’. However, I am not sure if I see any data from tissue DNA. Please clarify.

6. Did the authors consider using other body fluids, in particular vaginal secretions for this test? While serum/plasma may not carry markers of HPV infection in early invasive disease, vaginal fluids being local to the affected organ, could carry markers of HPV.

Minor comments

1.  Introduction could include a sentence or two about why improved diagnostics are needed for cervical cancers.

2. Please include DNA QC data in the supplemental

3. What is the amplicon length used for the library preparation? What was the sequencing depth?

4. Authors should include TMAP alignment modules/parameters used for alignment.

5. Have the authors made their work publicly available in online data repositories?

6. Why do the authors think the NGS assay is more sensitive? Especially as NGS sensitivity is dependent on sequencing depth. Depending upon the probe chemistry, qPCR could be more sensitive. What is the sequencing depth for the ion torrent NGS utilized in this study?

7. Authors mention in conclusions – ‘We have caried out a similar study (manuscript submitted simultaneously with the current study), which evaluated panHPV-detect in locally advanced cervical cancers, in patients undergoing chemoradiotherapy” –

Why have the authors published this as a separate study and not included it in a single manuscript?

Author Response

Thank you for your comments

Reviewer 2 Report

This article by Bryan et al, highlights an important aspect of harnessing the circulating tumor DNA to detect the cancer in early stage. In most of the cases, ability of current methods to capture the ctDNA varies with tissue where the tumor has developed.

This current work is a well design study and well written.I recommend for the article to be accepted.

Author Response

Thank you for your comments and taking the time to review this manuscript

Reviewer 3 Report

I read with great interest this Manuscript, which falls within the aim of the Journal.
Honestly, the topic is interesting enough to attract the readers’ attention. Nevertheless, the authors should clarify some points and improve the discussion by citing relevant and novel critical articles.

- OVERALL COMMENTS:

- English is fine, and there are minor typing errors to be revised

- I want to inform You that I make a plagiarism check routinely, and I can confirm that Yours is an original writing

-BACKGROUND

-Line 42-49. That information is too general. I suggest removing it to improve readability and focus on the topic.

- METHODS:

- To improve comparability in clinical practice and reproducibility of the results, I suggest including a little section with the cost of the sample calculation

-Because of the search of HPV, You should clarify if you included in the analysis any cervical histology for cervical cancer or just squamous and adenocarcinoma usual type. Please also specify if it has been considered during sample calculation.

- Also, clarify if you included HSIL, such as positive cytology or Hystopatological certain CIN 2-3.

- Please specify better inclusion criteria. All the affected patients seen during the study period were recruited, or there it was a selection? Was HPV vaccination an exclusion criteria? What about the previous infection?

- You should say wich staging system did you used are all IA1 and IB1 according to FIGO 2009 or FIGO 2018. It seems to be FIGO 2009. In this case, I suggest reformulating all the patients in FIGO 2018 staging.

- RESULTS:

- The calculated sample size still needs to be reached. Why did you interrupt the recruitment?

- To make it possible to understand the research, You should check for LVSI positivity and depth of infiltration

- The lack of FUP represents the more considerable bias. I suggest removing all the parts about FUP because it is an ancillary information but badly exposed

- DISCUSSION:
- The authors need to highlight their study's strengths and limitations adequately. I suggest better specifying these points
- What are the actual clinical implications of this study? it is essential to report the results obtained by the authors in the context of clinical practice and to adequately highlight what contribution this study adds to the literature already existing on the topic and to future study perspectives

- Critical Issues:

- Study design could be better. You should better include just patients with certain positive HPV cytology. The simple is few and made fewer by this bias. Otherwise, if the hypothesis is that HPV may reach blood steam, You must report the percentage of oncological patients with LVSI positivity and the depth of stromal infiltration. You should better reformulate all of Your analysis. The Matherial section, except for NGS explanation, should be revised entirely. Finally, the idea could be better developed by including a control arm of the patients with positive cytology for HPV but no histological lesion.

Author Response

Thank you for your comments - please see the attachment

Cost of sample calculation has been added to supplementary material 

Round 2

Reviewer 1 Report

I thank the authors for their responses to my comments. Authors have revised the manuscript adding more information which is helpful. However, there are critical issues that need addressing. Please find below my comments for the revised manuscript –

Major Points

1. The two paragraphs in ‘Methods’ from lines 140 to line 164 are exactly identical word for word with authors’ previously published work (https://doi.org/10.3390/cancers15051387). Authors cannot use the exact same verbiage between their published works as this constitutes self-plagiarism. Authors must paraphrase briefly and cite the original work where the method was first described in complete detail.

2. Regarding Library preparation optimizations and quality control of samples sequences –

2.1 Did the authors consider concentrating the samples to obtain higher amounts of DNA while still requiring less volume of library preparation reagents?

2.2 Can authors please include the cfDNA quantitation and quality data for each of the samples? Using lower amounts of DNA is only recommended when sample quality is very high. Otherwise, it could affect library yield and subsequently result in low coverage.

2.3 Did the authors have a minimum DNA quality cut-off set that each of the patient samples meet.

2.4 How much total cfDNA was obtained per patient? Is there a difference between total cfDNA amount (normalized to plasma volume) in the patient groups?

2.5 What percentage of the total reads generated (prior to filtering) align to the human genome?

2.6 Can authors include data to show library prep optimizations?

I could not find QC data in the supplemental.

3. Regarding assay sensitivity –

It is certainly beneficial to obtain data from different regions of the genome and different strains of HPV. However, this data can be lost if the abundance of the target is low, sample quality is low and/or sequencing depth is insufficient.

NGS results are typically validated by qPCR. Did the authors consider validating samples 8 and 10 from ‘Table 4’ for presence of HPV type 16 by qPCR?

Minor Comments

1. Regarding comparison of alignment –

My apologies. How does alignment vary with Bowtie vs Hisat vs BBMap? How would the alignment vary with quality cutoff set to 20 or 25?

2. Regarding using vaginal fluids –

I understand. However, these fluids can be self-collected, and many at-home self-test kits are in the market. Additionally, any patients that voluntarily come to the clinic for pap smear can also be recruited to the study, and extra samples collected during this routine test can be banked. Do authors have access to such a biobank? I highly recommend the authors to take this into consideration. 

3. What is the total number of reads per sample?

Is the average sequencing depth 70,000x per amplicon?

4. In line 238, authors state – “There was no correlation between the number of amplicons detected and the grade of CIN or stage of cancer.”

Authors should bear in mind that they do not have sufficient data/power to make an assessment regarding correlation between grade or stage and number of amplicons, as this would require enough N from each grade/stage

5. Authors should remain consistent in the naming of their two groups – ‘Early Invasive Cervical Cancer’ or ‘EICC’ and ‘Pre-invasive Cervical Disease/Lesion’ or PICD/PICL

The lack of consistency in the text and tables can be confusing for the reader.

I have attached annotated pdf and cover letter comments for better clarity with this review. 
